# Describing the Rehabilitation Workforce Capacity in the Public Sector of Three Rural Provinces in South Africa: A Cross-Sectional Study

**DOI:** 10.3390/ijerph191912176

**Published:** 2022-09-26

**Authors:** Thandi Conradie, Karina Berner, Quinette Louw

**Affiliations:** Department of Health and Rehabilitation Sciences, Faculty of Medicine and Health Sciences, Stellenbosch University, Cape Town 7505, South Africa

**Keywords:** rehabilitation, workforce, capacity, rural, public health

## Abstract

The World Health Organisation emphasises the importance of addressing gaps in health systems where rehabilitation services are poorly integrated. In South Africa, regions with the largest disability rates are usually the areas where rehabilitation is least accessible, due to inadequate workforce capacity. The first step towards addressing workforce capacity is to determine current capacity. This paper presents a cross-sectional study to describe rehabilitation workforce data in the public sector of three rural South African provinces. A web-based therapist survey and a telephonic facility survey was conducted. Data were collected regarding total number of therapists per province, age, years employed, qualifications, salary level, profession type, level of care, and rural/urban distribution. Descriptive statistics were conducted, while Chi-squared tests compared professions regarding highest qualification and salary level. Population-adjusted ratios were calculated using national uninsured population statistics. The web-based survey had 639 responses while the telephonic survey reported on 1166 therapists. Results indicated that the mean age of therapists across the respective provinces was 28, 35 and 31 years of age, and the mean employment years in the respective provinces were three, eight and five years. Most of the workforce (*n* = 574) had a bachelor’s degree as their highest qualification. A total of 27% of the workforce were community service therapists and 61% of therapists earned a production-level salary. Occupational therapy was best (40%) and speech and audiology therapy least (7%) represented. Three percent of therapists worked at primary level, versus eighty percent at secondary level. Forty percent of therapists worked in rural areas. Workforce density per province ranged from 0.71–0.98 per 10,000 population. Overall, results show that the rehabilitation workforce density is low, and that the distribution of therapists between rural and urban settings, and levels of care, is inequitable. Considering the rise in rehabilitation need, prioritisation and strengthening of the rehabilitation workforce capacity is vital to ensure integration across all levels of care and service.

## 1. Introduction

The *World Health Organisation (WHO) Rehabilitation 2030: A Call for Action* emphasises the need to address the gaps in health systems where rehabilitation services are not well integrated, ref. [1] to achieve Universal Health Coverage (UHC) [2]. This requires that health services are available at all levels of care and for all health needs, including health promotion, prevention, treatment, rehabilitation, and palliative care. Consistent with the United Nations Convention on the Rights of Persons with Disabilities (CRPD), rehabilitation services are vital throughout the lifespan, at all levels of care and for a wide range of conditions [1,3,4]. Rehabilitation has traditionally been presented as a ‘disability-specific’ service that caters for a small proportion of the population [5]. However, the CRPD recognises that disability is ‘an evolving concept’ and that the correlation of a person with impairments and their environmental or personal barriers can determine disability [4,6]. It is therefore important to consider that the need for rehabilitation is growing globally due to an ageing population and people living longer with increasing and chronic non-communicable diseases [5]. Thus, rehabilitation services must be scaled up significantly in both high-income countries (HICs) and low- and-middle-income countries (LMICs). However, upscaling poses a significant challenge in many LMICs as the rehabilitation workforce lacks the capacity to meet the demand for rehabilitation services [1,3].

The WHO developed the Rehabilitation in Health Systems: A Guide to Action (‘The Guide’) to assist countries or regions in assessing their baseline rehabilitation capacity [7]. The Guide suggests using the WHO’s six building blocks of the health system framework: leadership and governance, financing, service delivery, health information systems, medicines and technology, and health workforce [7,8], to better understand rehabilitation capacity. Without a sufficient workforce, none of the other components can function properly [3,9,10]. An equitably distributed rehabilitation workforce that is sufficient in number and skills is vital to achieving an efficient rehabilitation service [3,10]. Unfortunately, in many LMICs, where rehabilitation capacity structure and services are inadequate, the rehabilitation workforce is often left isolated with no support [3]. This often leads to poor service quality and poor workforce retention. The lack of health services and resources that has become apparent in each of these building blocks after the COVID-19 pandemic has shown, not only how unprepared rehabilitation services are for emergencies, but also the gaps in these services [11]. Notably, the component that has been affected most during this time was the rehabilitation workforce [12].

A rehabilitation workforce that is determined by a population’s needs—and thereby equitably distributed—is a vital component to achieving UHC [10,13]. In South Africa (SA), a mere 16% of the population are serviced by the private health sector, which makes up most of the SA health and rehabilitation workforce [10,13,14,15]. The remaining 84% of the population are serviced by the scarcely resourced public sector. Furthermore, regions with the largest disability rates are usually the areas where rehabilitation is least accessible [13]. The SA National Department of Health (NDoH) plans to achieve UHC by implementing National Health Insurance (NHI) [16]. Concerningly, rehabilitation remains excluded from most major health policies that are vital towards implementing NHI [10]. Although rehabilitation has been included in the new 2030 Human Resources for Health Strategy (the HRH Strategy)—from which it was previously excluded—there is no specific plan to strengthen rehabilitation capacity [15]. Furthermore, the data that are used to reflect the rehabilitation workforce capacity in the HRH Strategy are outdated and inaccurate [10,15]. There is therefore a need to provide updated evidence of the rehabilitation workforce capacity [10].

Identifying the gaps in rehabilitation workforce capacity data and providing baseline data will assist with the assessment of the rehabilitation workforce [7], particularly in under-served rural regions with reportedly poor access to rehabilitation services [10,13,14]. The Framework and Strategy for Disability and Rehabilitation Services (the FSDR) in SA reports an inequitable distribution of rehabilitation workforce and high vacancy rates across all levels of care, especially at primary care [13]. The FSDR also highlights the lack of reliable data of rehabilitation in the public sector, which limits planning of services and inclusion of rehabilitation into initiatives such as NHI [13]. Evidence-based information on rehabilitation workforce data cannot occur unless work is put into better availability, timeliness, comprehensive and reliable data [3]. The HRH Strategy recommends a workforce registry to provide reliable data that are immediately available to inform the planning and management of human resources for health [15]. The HRH Strategy cites the establishment of an electronic database and electronic health workforce registry as a means to capture important information and validate the status of the current health workforce in SA [15].

The first step towards improving rehabilitation workforce capacity data is to determine the current rehabilitation workforce capacity, especially in underserved, rural regions. This paper reports on a cross-sectional study to describe the rehabilitation workforce data on the number of therapists, distribution by population, type of therapists, qualification level, and distribution between rural and urban settings, salary levels and levels of care, in the public sector of three rural provinces in SA.

## 2. Materials and Methods

### 2.1. Study Design

A cross-sectional study using a web-based and telephonic survey was conducted between 2020 and 2021. The study was commenced in March 2020 but was halted due to lockdowns and institutional restrictions which barred the continuation of non-COVID related research for at least 6 months in 2020.

We designed a web-based survey using a Research Electronic Database Capture (REDCap) form for therapists to provide detailed information (see Table 1). This web-based survey was complemented by a telephonic facility survey which involved contacting each facility’s manager to collect data on the rehabilitation workforce (See Table 1). Table 1 shows the descriptors that were respectively included in the two surveys. The descriptors were selected based on consultation with key stakeholders such as governmental officials and The Guide mentioned above [7].

This study is a sub-study of a project on assessing the capacity of rehabilitation in SA according to WHO’s recommendations in The Guide. This project has been conducted in partnership with the NDoH and WHO’s SA office. A concept note was developed at the start of the project with the NDoH who have provided support throughout this study and the capacity assessment project.

REDCap is a secure, web-based software platform designed to support data capture for research studies hosted at Stellenbosch University [18,19]. REDCap has a two-step verification process to ensure security the of data. The survey was designed in a format that allowed participants to enter personal information whilst maintaining confidentiality during data analysis. The REDCap application is therefore secure and has 128-bit encryption between data entry and the server. Although REDCap is designed and maintained by Vanderbilt University, the data and application are stored on servers provided by Stellenbosch University Information and Communications Technology Division. The personal information stored online can only be accessed by the researcher and the software engineer, each needing to follow a two-step verification process to access the REDCap projects, i.e., sign-in with Stellenbosch University username and password, followed by entering a code using Google Authenticator.

Ethical approval was obtained from the Stellenbosch University Health Research Ethics Committee (N19/04/048). Permissions were obtained from participating provincial Departments of Health (DoH) and the provincial rehabilitation managers. All participants granted voluntary informed consent to participate in the study.

### 2.2. Setting

The study was conducted in three of SA’s rural provinces, referred to as P1, P2 and P3. These three provinces have the highest rural population rates [20] and the highest poverty levels [21] in SA compared to the other six provinces. P1 has the largest population of the three provinces (the third highest in SA), while P2 and P3 have the fifth and sixth highest, respectively [22]. According to the most recent (2016) general household survey, P3 has the lowest health expenditure, with P1 and P2 the third and fourth lowest, respectively [23].

### 2.3. Eligibility Criteria

The study population was limited to the rehabilitation workforce working at all levels of care (primary, secondary and tertiary) in SA’s DoH public sector. These included audiologists, dually qualified speech and audiology therapists (SAT), occupational therapists (OT), physiotherapists, speech and language therapists (SLT), and physiotherapy or OT assistants and technicians (see Appendix A for definitions of profession type), and rehabilitation facility managers. Recently graduated therapists were also included because they are mostly placed at rural facilities to complete one year of public service (community service) before they can obtain full registration from the Health Professions Council of South Africa (HPCSA). Rehabilitation workers employed by the Department of Education were excluded in this study as they do not report to the DoH.

Recruitment was carried out via existing DoH communication channels in the provinces (WhatsApp groups and memorandums sent to clinical managers of each facility, email communication), social media platforms (Facebook, Instagram, and Twitter), and personal communication via the DoH rehabilitation and district managers in the participating provinces.

### 2.4. Data Collection Procedures

#### 2.4.1. Preparing for Data Collection

The three provincial DoH rehabilitation managers were consulted regarding the most feasible communication strategies to reach therapists (for the web-based survey) and facility managers (for the telephonic survey). One week before commencing data collection in each province, the research team used short video clips via the suggested communication channels to explain the project aims, rationale and how the therapists should complete the survey.

#### 2.4.2. Data Collection

##### Web-Based survey

Data were collected and managed via the web-based survey (tailored REDCap form) or in PDF format. The survey, accessed via a web link, consisted of two screens. The first screen consisted of seven data fields including informed consent details and personal details. The second screen consisted of 50 data fields which related to employment demographics. Adaptive questioning (e.g., branching logic) was added so that participants were required to complete only 12 data fields.

##### Telephonic Survey

Not all therapists completed the web-based survey. Thus, the telephonic survey was used to obtain accurate data on the number and level of therapists at each facility in the province. The current national public system’s staff database (PERSAL) could not be used due to inaccuracies of the data—the number of posts are often reflected inaccurately or other health professionals are appointed in therapists’ positions [13]. The telephonic facility survey consisted of informed consent and eight questions regarding the therapists working at the facility.

#### 2.4.3. Data Validation

At least one stakeholder meeting was held with facility managers (who were familiar with the rehabilitation workforce information at their facility), district managers, provincial managers, and participating therapists to verify the collected rehabilitation workforce data.

To encourage therapists to attend the feedback session, therapists could obtain continuous professional education points for attending the session which provided information and background on the project. The first data validation meeting was face-to-face, but due to COVID-19 restrictions, subsequent validation meetings were conducted online. Figure 1 presents the data collection timeline.

#### 2.4.4. Data Analysis

Data, which excluded personal identifiers, were downloaded from REDCap (web-based therapist survey) into MS Excel and cleaned for analysis. REDCap allows the online creation of reports, where one can select the desired data to download. This allows the researcher to only download relevant data and all personal information can therefore be omitted. Descriptive statistics were calculated and presented using tables and graphs for visualisation. Population ratios (for provinces and districts) were presented using population statistics as reported by Statistics of SA [22]. The ratio was calculated by dividing the number of therapists by the national uninsured population and multiplying by 10,000. To compare profession types regarding highest level of qualification and salary level, chi-squared tests were performed in SPSS Version 23.

## 3. Results

The response rate of the web-based therapist survey (i.e., the percentage of therapists who completed the web-based therapist survey in relation to the actual number of therapists based on the telephonic facility survey) was 55% (*n* = 639/1166). In P1 the response rate was 76% (*n* = 286/377). P2 had the lowest participation rate of 22% (*n* = 100/454). P3 had a participation rate of 71% (*n* = 253/355).

### 3.1. Web-Based Therapist Survey

#### 3.1.1. Age

Table 2 shows the ages (mean [SD] and range) of the therapists per province and per profession. Ages ranged between 22–64 years across all three provinces. Between provinces, P2 had the highest mean age (34.47 years) and between profession types, OT had the highest mean age (31.71 years). The youngest age is 22 years in both P1 and P3, and in all the professions.

#### 3.1.2. Years at Current Facility

Table 3 shows the number of years that therapists worked at their current facility. The shortest duration was less than a month in P1 and the longest duration was 30.25 years in P3 and OT. Between provinces, P2 had the highest mean years (7.88 years) and between professions, SAT had the highest mean years (8.47 years).

#### 3.1.3. Highest Level of Qualification

There was a significant difference (*p* < 0.000) in the highest qualification between professions (see Table 4). OT had the highest percentage of postgraduate degrees (41%; *n* = 16/39). OT also had the highest percentage of master’s degrees (41%; *n* = 15/37). Only one physiotherapist had a doctoral degree, and one OT had a postgraduate diploma.

#### 3.1.4. Salary Levels

The total percentage of therapists (all professions combined) per salary level and across all three provinces are shown in Figure 2. Most therapists (61%) were classified as earning production level salaries and almost a third (27%) of all therapists were community service therapists.

Figure 3 shows the salary level percentages per profession type, for all three provinces combined. There was a significant difference (*p* < 0.000) in the salary levels between professions. The largest percentage of community service therapists was observed within the SLT profession (48%), and the lowest within the SAT profession (15%).

### 3.2. Telephonic Facility Survey

#### 3.2.1. Profession Type

The percentage of professions for each of the three provinces based on the telephonic facility survey (total *n* = 1166) is shown in Figure 4. The OTs contributed the largest number (*n* = 252; 40%) of all therapists in the three provinces. SAT contributed the smallest number of therapists (*n* = 47; 7%).

The distribution of professions per province was found to be similar between the three provinces (see Figure 5). Physiotherapy had the largest number in P1 compared to P2 and P3 where OTs were more.

#### 3.2.2. Level of Care

The percentage of professions per level of care from the telephonic facility survey is depicted in Figure 6. There were no SATs at primary level but were present at tertiary level (*n* = 12) and secondary level (*n* = 51). At tertiary level, physiotherapy (*n* = 76) had the largest number of therapists and at primary care OT have the highest number (*n* = 18). Many of the facilities at secondary level conducted outreach to primary care facilities.

Figure 7 shows the distribution of therapists per level of care. Only 3% of the therapists worked at primary care in comparison to the 80% working at secondary level.

#### 3.2.3. Rural Versus Urban

Figure 8 shows the telephonic facility survey distribution of the number of therapists working in rural and urban facilities per province. The percentage of therapists working in rural facilities was 40% (*n* = 470) and the other 60% (*n* = 696) of the therapists worked in urban facilities. Different distribution when comparing urban and rural were noted between the provinces. In P1, 16% (*n* = 60) versus 57% (*n* = 257) of the therapists in P2 worked in rural facilities. In P3, the number of therapists working in rural areas was less than in urban areas by 8% (*n* = 29).

#### 3.2.4. Distribution per Uninsured Population

The uninsured population (80%) was calculated using the data provided in the 2016 Community Survey [22]. Figure 9 shows the telephonic facility survey profession number per 10,000 uninsured population. OT in P2 had the highest population adjusted ratio at 0.46 per 10,000. SAT in P1 and audiology in P2 had the lowest ratios at 0.02 per 10,000.

The highest number of profession type per 10,000 uninsured population per provincial district was OT at 0.68 in district nine, while the lowest number of therapists is 0.01 for audiology in district one, two and eight and SLT in district two (see Figure 10).

The districts with the higher populations tended to have a larger number of therapists (see Table 5). For example, district nine in P2 with over one million people, had 12.54% (*n* = 149) therapists.

Table 6 shows the number of therapists per province in relation to the national uninsured population. Overall (all therapists combined), P1 had a ratio of 0.71/10,000 population, P2 had a ratio of 0.98/10,000 and P3 a ratio of 0.97/10,000 population.

## 4. Discussion

We described the rehabilitation workforce in the public sector of three rural SA provinces. Overall, the rehabilitation workforce capacity of the three provinces is severely lacking in quantity and have large disparities in the workforce distribution between levels of care and across the urban/rural divide. The bulk of the workforce comprised of production level and community service therapists, the rehabilitation workforce was relatively young (high twenties to mid-thirties) and on average, therapists were employed for less than nine years at their current facilities.

Population adjusted ratios are routinely reported as a metric to describe the rehabilitation workforce density [24,25,26,27]. This study showed alarmingly low population-adjusted ratios of the total number of therapists in each of the three provinces per uninsured population (see Table 6). There were no notable between-province differences since the ratio was less than one therapist per 10,000 population in all three provinces (irrespective of type of therapist). This implies that people requiring rehabilitation may not have access to rehabilitation services [24]. The accurate number of people with disability in South Africa is unknown, but is estimated to be about 8% (9%, 6% and 8% in the three provinces, respectively) [21]. Persons with disability require long-term, intensive rehabilitation as well as transport to the closest facility, which is often at primary care, where the rehabilitation workforce is scarce. The limited access contradicts the CRPD. However, rehabilitation services are not limited to a select few people who require disability-specific services [5], as many people living with chronic disease or recovering from an injury may also require rehabilitation services at any given point in their life span. Thus, our findings indicate that the distribution of the workforce (based on total therapists) is arguably inadequate to meet the population needs.

We also presented the distribution by type of therapists per population. Two studies conducted in SA on the profile of OTs [10], and SLTs, SATs and audiologists [14] demonstrated double the ratio compared to what we found in the telephonic facility survey; albeit less than one per 10,000 of the population. The differences in findings may be explained by the methodological differences in data sources used and the calculation of the ratios. Ned et al. (2020) and Pillay et al. (2020) used data from the HPCSA to determine the number of therapists and used the PERSAL data to calculate the distribution between the public and private sectors [10,14]. The total population was used to calculate the ratios and the percentages were not adjusted to the uninsured population. Both studies reported that most of the workforce was employed in the private sector but did not adjust the population ratios accordingly. The ratios from the current study for physiotherapy and SLT, SAT and audiology (combined) were comparable to the HRH strategy [15], which is based on PERSAL data. For OT, a slightly higher ratio is noted (0.35 compared to 0.26 per 10,000 in the HRH Strategy). However, despite the slight variances in findings, the OT and SLT, SAT and audiology studies, the HRH Strategy, and the current study concur that the therapist ratios to the population are very low in SA.

Despite being an upper middle-income country, the ratio of therapists to population in SA compares to lower middle-income countries such as Bangladesh (less than 0.1 physiotherapists and occupational therapists per 10,000) [24]. The physiotherapy population ratios in some HICs such as Portugal [24], the United States of America [24] and Ireland [28] were almost 10 times higher than the physiotherapy population ratios found in this study. The NDoH has excluded rehabilitation in the health planning and therefore have fewer financial resources to retain or create new posts for therapists [13]. Rehabilitation services are often the lowest priority when financing for human resources for health are considered, [12,29] creating a double burden for poorly resourced provinces as the budgeting of posts are resourced at a provincial level. Once funds become available, infectious diseases are usually prioritised [29]. Resultantly, vacant rehabilitation posts are frozen, and therapists are forced to seek employment in the private sector or overseas [10]. Similar trends have been noted in other upper middle-income countries such as Brazil [26,30]. These low ratios show disparities in the rehabilitation workforce at all levels of care.

The number of therapists were extremely low in primary care (see Figure 5 and Figure 6). Similar maldistribution of therapists between levels of care were reported in Brazil [30]. In contrast, some HICs have focused on improving access to primary care rehabilitation, for example, Ireland has 74% of the physiotherapists working at community level [28]. The services available at the hospital level are often very limited, people being discharged with no follow-up and often without assistive devices. Most of the population who rely on primary care services are already disadvantaged in their access to healthcare due to lack of transport and increased out-of-pocket expenses [31,32]. Thus, the populations that have a larger need for rehabilitation services are often those that have the least access to health facilities with increasingly negative knock-on effects for the most vulnerable persons in our population. This large disparity between levels of care with low numbers at primary care, are mostly located in rural areas.

An inequitable distribution of therapists was found between urban and rural facilities with more than half of the rehabilitation workforce being situated in urban areas (see Figure 7). The recruitment and retention of healthcare professionals is a global problem [33]. Several studies on human resources in other countries show a large disparity between rural and urban areas in both HICs and LMICs [34,35,36]. In SA, most of the rural hospitals are in what previously used to be homelands during the apartheid era. Each homeland had its own health department and professional bodies [37]. When apartheid ended, the management of these hospitals was taken over by the national government. As these hospitals were far from the cities and universities, they were often neglected during government’s planning and resources. Due to the poor resources at these hospitals, very few healthcare professionals are attracted to work in these areas and if they do, the working and living circumstances are often too poor to retain staff [38]. Additionally, undergraduate training does not prepare therapists for the high workload and case-mix seen at rural hospitals [13]. Despite this longstanding disparity between urban and rural rehabilitation workforce, strategies to rectify this maldistribution have only recently been included in the HRH Strategy.

The NDoH have attempted various strategies such as a compulsory community service year and rural allowance to improve recruitment and retention to rural areas. This has improved the staffing crisis in some areas in the short-term. Khan et al. (2009) reported the willingness of therapists to work in rural areas beyond their community service year when they begin but changed their minds by year-end [39]. Many therapists reported poor work satisfaction and motivation resulting from feeling isolated professionally and socially and having little or no supervision available. Therapists have reported that the financial incentive is often not enough to motivate longer services at rural facilities. The implemented strategies have not effectively improved the recruitment and retention of therapists in these areas.

Newly qualified therapists are thrown into the deep end and expected to make a difference with minimal or non-existing supervision and support from other professions or management. The mean ages from our study (see Table 2) show that the rehabilitation workforce, in general, is young and have minimal experience. The mean years working at their current facility (see Table 3) corroborate these findings as therapists often do not stay on at public facilities in the long term. This leads to a high turnover rate at hospitals causing disruption in continuity of services due to lack of “institutional memory” [40]. Young therapists often report that there is little opportunity for career advancement in many public sector facilities where there is no supervision. Similar reasons were reported by young physiotherapists in Canada who felt that they lacked opportunities for professional development [35]. Only a few of the participants in the current study reported completing a postgraduate degree. A study conducted amongst private sector physiotherapists in SA cited reasons including lack of time to do a postgraduate course whilst working as there is often no study leave and expensive costs of studying full-time [41]. This parallels the situation in the public sector where there is often no study leave and no financial incentive for career advancement.

The rehabilitation workforce capacity of the three provinces included in the current study is severely lacking in quantity and have large disparities in the distribution of the workforce between levels of care and across the urban/rural divide. This description of the rehabilitation workforce can be generalised to other rural provinces and are comparable to national figures in SA. Although SA is an upper middle-income country, the rehabilitation workforce capacity is similar to the capacity in LMICs. This is extremely concerning as minimal rehabilitation workforce are available in the areas with the most vulnerable populations. With an increase in rehabilitation need and an inequitable distribution of the rehabilitation workforce, the future of rehabilitation services in the public sector does not look very promising for the most vulnerable persons in our population.

## 5. Limitations

About 40% of the therapists did not complete the web-based therapist survey. Therefore, the data on qualification, age, salary level and years worked at their current facility was limited to only those who completed the web-based therapist survey, and this could have introduced bias. The study was limited to quantitative data and did not include the quality or efficiency of the rehabilitation services. This was a cross-sectional study and therefore we were unable to describe the trends over time and the impact of COVID-19 on the rehabilitation workforce capacity. We collected data over one year, as the data collection was disrupted by the COVID-19 restrictions. The data collection in P1 was completed before COVID-19, while the data collection in P2 and P3 were conducted once lockdown levels were eased. This increased time period between data collection in the provinces may have had an impact on the change of the data due to COVID-19 which was not accounted for in this study. The study included three rural provinces and therefore we were unable to compare the data between rural and urban provinces. The data collected was limited to the public sector and therefore excluded the workforce in other sectors such as the private sector or non-governmental organisations. Although it would have been useful to interpret the workforce distribution in terms of the number of people with disabilities in SA, accurate data to base such a discussion on are lacking—i.e., data regarding the precise number and distribution of persons with disability in SA. This lack of data is a result of a major limitation in determining true levels of disability in SA, namely that disability prevalence surveys are usually based on reported disability—often by a proxy informant—which may over- or underestimate the prevalence [13]. There is currently no minimum value of number of rehabilitation workers globally and nationally. Our study results could therefore not be interpreted against any existing published minimum standards on quantity per population (given the lack of such published data).

## 6. Recommendations

As this study was only conducted in three provinces and the overall response rate of the web-based therapist survey was about 40%, not all the demographics of the therapists could be validated. Primary and longitudinal studies should therefore be conducted to develop strategies to enable reliable real-time data. The mean age of the rehabilitation workforce and the percentage of community service therapists suggests that the workforce is young inexperienced. As this is a quantitative study, the quality of the services and competency of the workforce cannot be determined. Qualitative studies should be conducted which aim to understand more about the quality of rehabilitation services, the needs of the population and the competencies of the rehabilitation workforce. Rehabilitation has been excluded from most major policies and guidelines and are therefore not integrated at all levels of care and in rural areas. Research on the cost of rehabilitation services and the rehabilitation workforce should be conducted to assist policy makers and the NDoH in including rehabilitation in policies and planning (especially budget planning). The results from this study included only rehabilitation workers in the public health sector. To obtain a clearer understanding of the overall rehabilitation workforce capacity in SA, future research should include the private health sector and other ministries such as the department of education. Further research on a rehabilitation workforce capacity building should also be conducted. This research should be conducted in collaboration with the NDoH involving governance and should include data on policies, disability and rehabilitation, and social welfare.

## 7. Conclusions

The aim of this study was to describe the rehabilitation workforce in the public sector of three rural provinces in SA. The rehabilitation workforce was described according to the age and number of therapists, years worked at facility, the distribution by population, the type of therapists and qualifications, the distribution between urban and rural, salary levels and levels of care. This study found that there is a low density of rehabilitation workforce and an inequitable distribution of therapists between rural and urban and levels of care, with the lowest number at rural and primary care. The rehabilitation workforce working in these three provinces is relatively young and inexperienced. This means that the population with the greatest need for rehabilitation has the poorest access to rehabilitation services. With the increasing need for rehabilitation, it is vital that the rehabilitation workforce capacity is strengthened and prioritised, ensuring integration at all levels of care and services.

## Figures and Tables

**Figure 1 ijerph-19-12176-f001:**
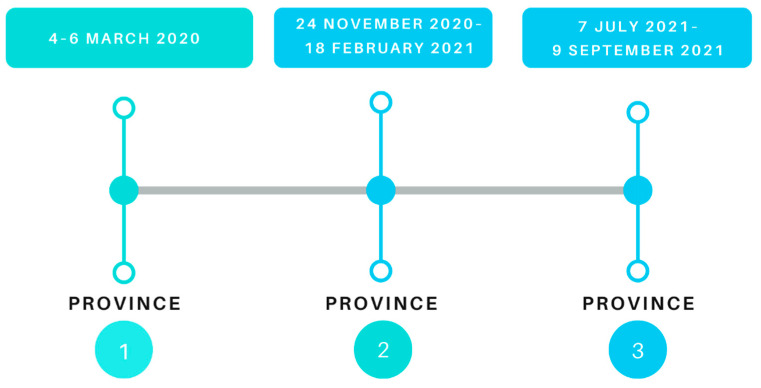
Timeline of data collection.

**Figure 2 ijerph-19-12176-f002:**
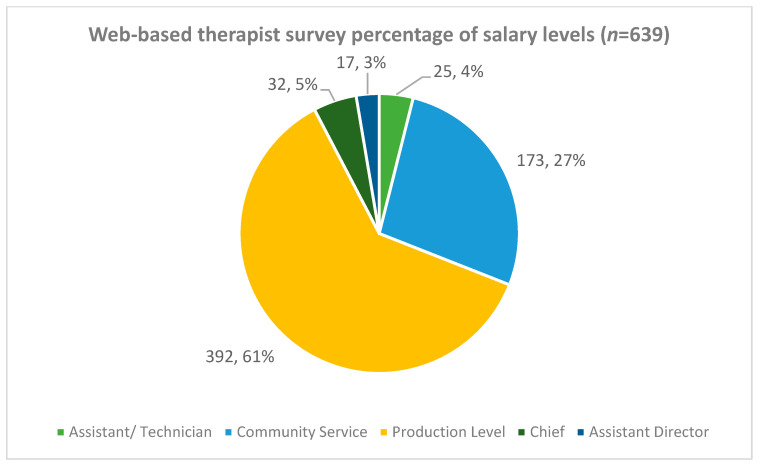
Percentages of rehabilitation therapists (all types combined) within different salary levels across the three provinces.

**Figure 3 ijerph-19-12176-f003:**
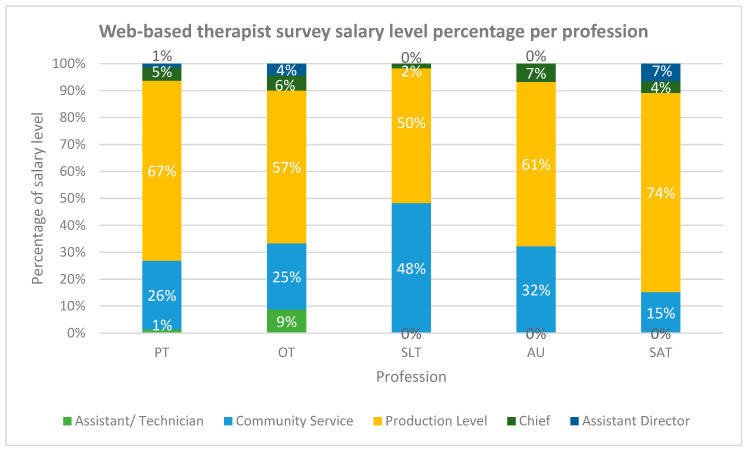
Distribution of salary levels per profession type. Legend: PT—physiotherapy, OT—occupational therapy, SLT—speech–language therapy, AU—audiology and SAT—speech and audiology therapy.

**Figure 4 ijerph-19-12176-f004:**
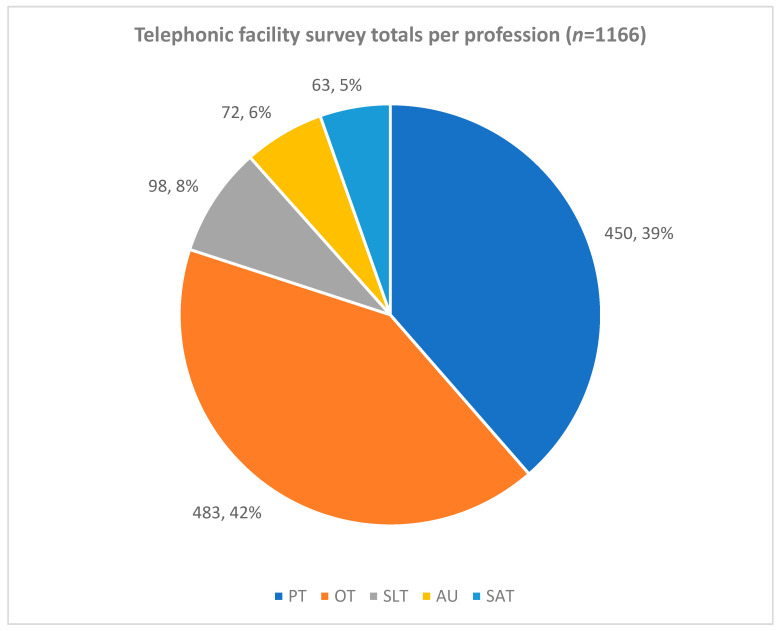
Total therapists based on the telephonic facility survey total number of three provinces and percentage of profession type (*n*;%). Legend: PT—physiotherapy, OT—occupational therapy, SLT—speech–language therapy, AU—audiology and SAT—speech and audiology therapy.

**Figure 5 ijerph-19-12176-f005:**
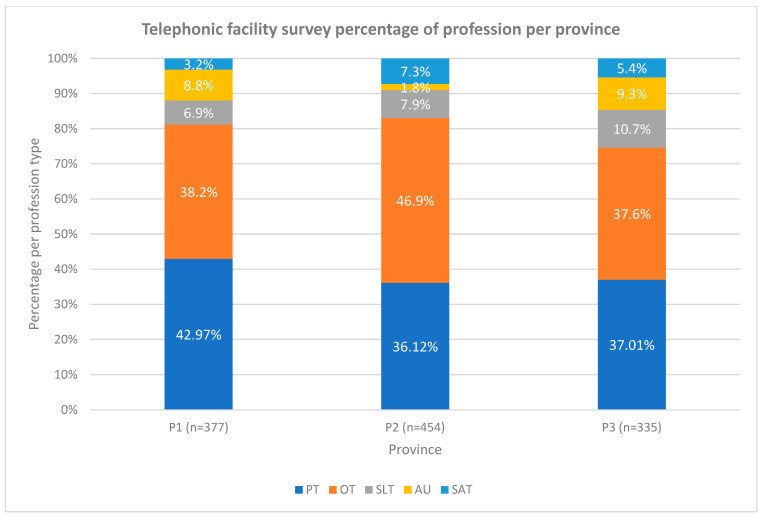
Distribution of the telephonic facility survey of profession per province. Legend: PT—physiotherapy, OT—occupational therapy, SLT—speech–language therapy, AU—audiology, SAT—speech and audiology therapy and P—province.

**Figure 6 ijerph-19-12176-f006:**
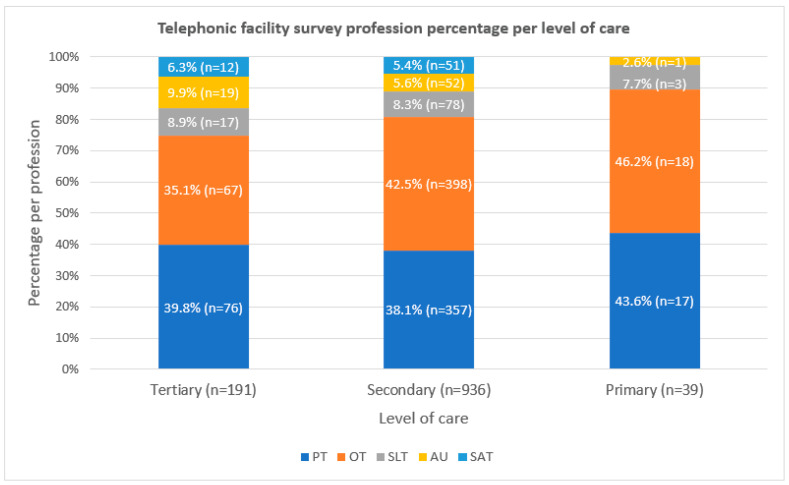
Telephonic facility survey distribution of profession type percentage per level of care. Legend: PT—physiotherapy, OT—occupational therapy, SLT—speech–language therapy, AU—audiology and SAT—speech and audiology therapy.

**Figure 7 ijerph-19-12176-f007:**
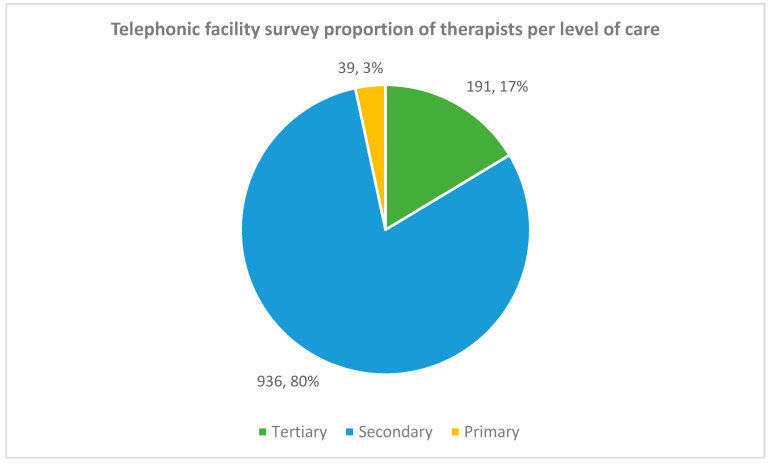
Telephonic facility survey of therapists per level of care.

**Figure 8 ijerph-19-12176-f008:**
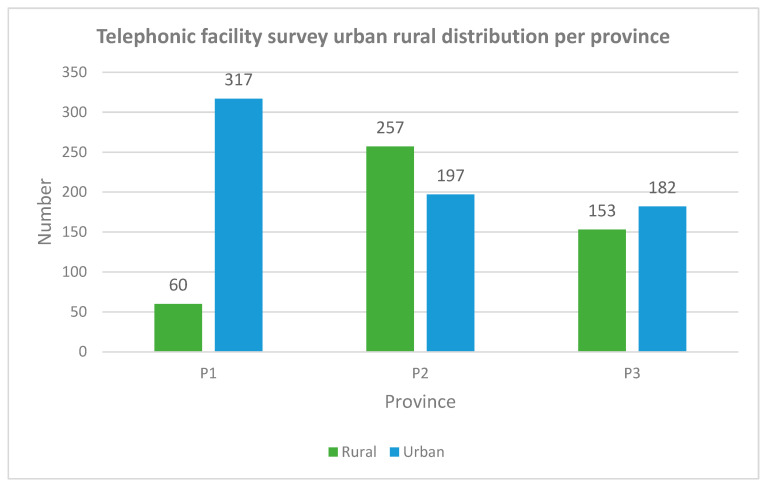
Distribution of profession types per rural/urban area. Legend: P—province.

**Figure 9 ijerph-19-12176-f009:**
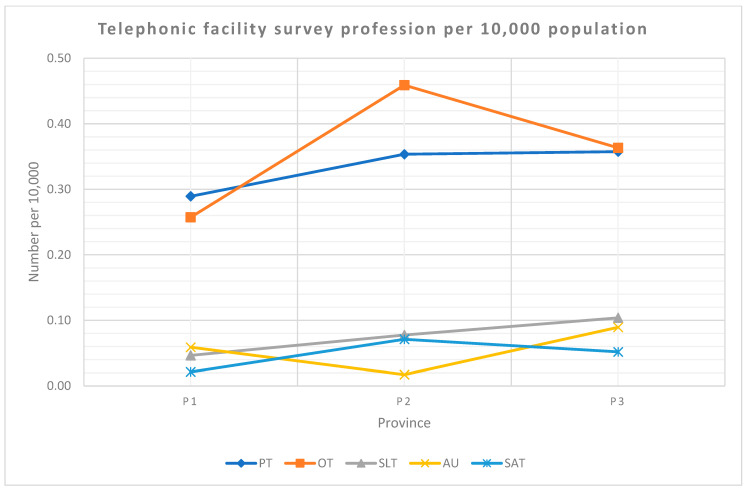
Telephonic facility survey profession distribution per 10,000 population per province. Legend: PT—physiotherapy, OT—occupational therapy, SLT—speech–language therapy, AU—audiology, SAT—speech and audiology therapy and P—province.

**Figure 10 ijerph-19-12176-f010:**
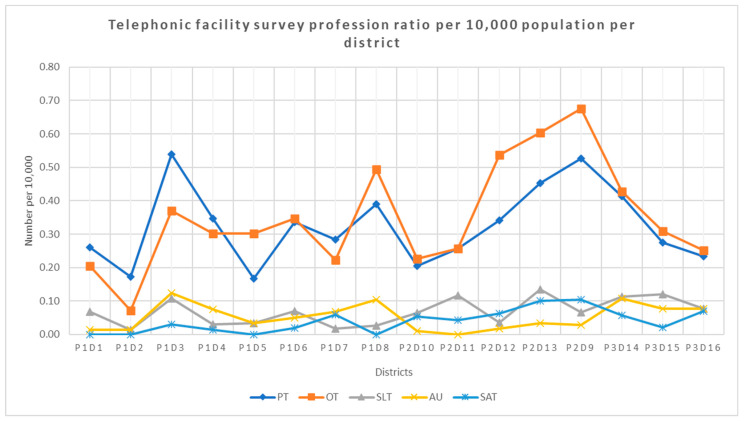
Telephonic facility survey profession distribution per 10,000 population per district. Legend: PT—physiotherapy, OT—occupational therapy, SLT—speech–language therapy, AU—audiology, SAT—speech and audiology therapy and P—province and D—district.

**Table 1 ijerph-19-12176-t001:** Table 1 lists the studied variables; definitions of these are included in Appendix A.

Online Survey (Web-Based Therapist Survey)	Telephonic Survey (Telephonic Facility Survey)
Age of therapists	Total number of therapists and profession type ^1^ Audiology Occupational therapy Physiotherapy Speech and language therapy Speech and audiology therapy
Total years employed at facility	Level of care ^1^ Primary Secondary Tertiary
Highest level of qualification	Rural ^2^ versus urban ^1^
Salary level ^1^ Assistant/Technician Community service Production level Chief Assistant Director	

^1^ See Appendix A for definitions. ^2^ Rural is defined as “Sparsely populated areas in which people farm or depend on natural resources, including villages and small towns that are dispersed through these areas [17].”.

**Table 2 ijerph-19-12176-t002:** Therapist ages according to province and profession, as derived from the web-based survey.

	Mean Age (SD)	Range	
		Lowest	Highest
Province			
1	27.82 (6.43)	22	59
2	34.47 (7.63)	23	55
3	30.87 (9.33)	22	64
Profession			
PT	29.35 (6.84)	22	56
OT	31.71 (10.02)	22	64
SLT	26.68 (4.60)	22	44
AU	27.9 (6.20)	22	59
SAT	31.6 (6.82)	22	51

Legend: PT—physiotherapy, OT—occupational therapy, SLT—speech–language therapy, AU—audiology, SAT—speech and audiology therapy.

**Table 3 ijerph-19-12176-t003:** Years that therapists worked at their current facilities according to province and profession, as derived from the web-based therapist survey.

	Mean Years (SD)	Range	
		Lowest Value	Highest Value
Province			
1	3.08 (7.82)	0.00	14.33
2	7.88 (5.18)	0.58	23.58
3	4.79 (6.50)	0.00	30.25
Profession			
PT	3.8 (4.53)	0.08	26.17
OT	5.40 (6.75)	0.08	30.25
SLT	2.35 (3.18)	0.08	12.50
AU	2.85 (3.40)	0.17	9.75
SAT	8.47 (17.51)	0.17	21.17

Legend: PT—physiotherapy, OT—occupational therapy, SLT—speech–language therapy, AU—audiology, SAT—speech and audiology therapy.

**Table 4 ijerph-19-12176-t004:** Highest level of qualification per profession as derived from the web-based therapist survey, reported as count and percentage within each qualification.

Qualification		PT	OT	SLT	AU	SAT	Total Count
*Certificate*	Count	0	4	0	0	0	4
	Percentage	0.0%	100.0%	0.0%	0.0%	0.0%	
*Diploma*	Count	1	20	0	0	0	21
	Percentage	4.8%	95.2%	0.0%	0.0%	0.0%	
*Bachelors*	Count	215	212	51	54	42	574
	Percentage	37.5%	36.9%	8.9%	9.4%	7.3%	
*PG Diploma*	Count	0	1	0	0	0	1
	Percentage	0.0%	100.0%	0.0%	0.0%	0.0%	
*Masters*	Count	6	15	7	5	4	37
	Percentage	16.2%	40.5%	18.9%	13.5%	10.8%	
*Doctorate*	Count	1	0	0	0	0	1
	Percentage	100.0%	0.0%	0.0%	0.0%	0.0%	
*Other* ^1^	Count	0	0	0	0	1	1
	Percentage	0.0%	0.0%	0.0%	0.0%	100%	

Legend: PT—physiotherapy, OT—occupational therapy, SLT—speech–language therapy, AU—audiology, SAT—speech and audiology therapy and PG—postgraduate. ^1^ Other—participant added incorrect date of qualification and specified degree as ‘other’.

**Table 5 ijerph-19-12176-t005:** Telephonic facility survey district details.

District	Total Number of Therapists	Percentage per District	Uninsured Population
P1D5	16	1.37%	298,672
P1D8	39	3.34%	383,938.40
P2D13	79	6.78%	596,606.40
P1D3	76	6.52%	648,422.40
P1D4	51	4.37%	664,395.20
P1D2	19	1.63%	694,314.40
P1D1	40	3.43%	731,856
P3D15	73	6.26%	908,327.20
P2D10	52	4.46%	927,348
P2D11	63	5.40%	935,809.60
P1D6	83	7.12%	1,010,440.80
P2D9	149	12.78%	1,064,348.80
P2D12	111	9.52%	1,115,159.20
P3D16	82	7.03%	1,156,499.20
P1D7	76	6.52%	1,165,541.60
P3D14	157	13.46%	1,403,944.80

Legend: P—province and D—district.

**Table 6 ijerph-19-12176-t006:** Telephonic facility survey therapists per national uninsured population.

Province	Total Number of Therapists	Ratio	Uninsured Population
P1	377	0.71	5,597,580.8
P2	454	0.98	4,639,272
P3	335	0.97	3,468,771.2

## Data Availability

Not applicable.

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
