# Peer review of "Describing the Rehabilitation Workforce Capacity in the Public Sector of Three Rural Provinces in South Africa: A Cross-Sectional Study"

_ijerph, 2022, doi:10.3390/ijerph191912176_

Round 1
Reviewer 1 Report
Dear Authors
Hope you are well.
I find your study very interesting and very well done, rehabilitation is an important health strategy. Due to the lack of rehabilitation professionals and lack of rehabilitation services, the need to strengthen rehabilitation is a pertinent theme. My suggestion is to consider in the future, therefore at the recommendations section, the possibility of a workforce capacity building project (that you may apply for financing). It is important, also to clarify the involvement of governments in this mission, to define clear terms of reference and to use a systematic pathway for the situation assessment. Information must be collected regarding policies, health, disability, rehabilitation, social security systems. Governance, political will and a common understanding of rehabilitation are crucial for implementation of the process. I think this should be said in the final sections.
Another think call for my attention, the ethical issues in you research are not contemplated, you should describe what kind of ethical procedures have been taken into account, including secrecy, confidentiality, data protection and data security, where data are stored etc.
Best regards
Reviewer 2 Report
This work presents a quantitative study about the rehabilitation workforce in the public sector of three provinces in South Africa. The manuscript is well-written, the method is clear, and the results understandable. Nevertheless, the study is descriptive about the situation of three South Africa provinces and its implications for future researches or its interest for readers is not enough. The data compiled and provided is more informative than of interest for others. The conclusions cannot be applied at all, nor for other contexts or countries. The limitations are severe but well-identified by the authors.
Additionaly, there are some mistakes (list not exhaustive):
1. line 100, acronym RW is not defined.
2. line 103 and 125: it is not Appendix 1 but A
3. Table 4 is not near to its title, confusing table 3 heading
4. Line 213. Wrong heading
5. Figure 3 and 5: STA > SAT
6. Table 5 would be better ordered by uninsured population
Reviewer 3 Report
1- It is stated in the title that the purpose of the study is to describe the data related to the rehabilitation workforce in South Africa, while the entire text of the article talks about the description of the rehabilitation workforce. It is suggested that the word "data" be removed from the title.
2- The number of participants in the study is not clearly stated in the article.
3- The title of table 3 tells about the years of work of the participants in the study, but the content of the table shows their number according to their educational rank.
4- According to the information provided in Table 3, the number of participants in the study is 638, while in Figure 2, this number is 639.
5- A comparison table about the distribution of the rehabilitation workforce in the three provinces of South Africa has not been prepared. It is suggested to compare the labor force distribution in three provinces and discuss it in the discussion section.
6- Such a study needs to provide accurate and clear information about the number of disabled people in the three provinces of South Africa, so that based on the information, we can talk about the distribution of the rehabilitation workforce.
7- In general, the research design is weak in terms of being able to provide a valuable message about the condition of disabled people who need services of the rehabilitation workforce. Introducing the UN's Convention on the Rights of Persons with Disabilities (CRPD) in the introduction section enriches the content of the article.
Reviewer 4 Report
1. I suggest making the font size (particularly those percentage numbers) in figures. The authors can try to print out their manuscript to see the difficulty of reading the figures.
2. The bullet points in sections 5 and 6 appear a bit weird. I would suggest choosing a different presentation.
Round 2
Reviewer 3 Report
Recommendations 5 to 7 expressed in the previous review, were not implemented by the authors. These recommendations are very important in terms of enriching the article, so if it is not possible to implement those recommendations, in my opinion, the article is not worth publishing.
